# Multifaceted Roles of GLP-1 and Its Analogs: A Review on Molecular Mechanisms with a Cardiotherapeutic Perspective

**DOI:** 10.3390/ph16060836

**Published:** 2023-06-03

**Authors:** Sudhir Pandey, Supachoke Mangmool, Warisara Parichatikanond

**Affiliations:** 1Department of Pharmacology, Faculty of Pharmacy, Mahidol University, Bangkok 10400, Thailand; sudhir.pan@mahidol.ac.th; 2Department of Pharmacology, Faculty of Science, Mahidol University, Bangkok 10400, Thailand; supachoke.man@mahidol.ac.th

**Keywords:** glucagon-like peptide-1 (GLP-1), GLP-1 receptor (GLP-1R), hyperglycemic cardiomyopathy, cardioprotective effects

## Abstract

Diabetes is one of the chronic metabolic disorders which poses a multitude of life-debilitating challenges, including cardiac muscle impairment, which eventually results in heart failure. The incretin hormone glucagon-like peptide-1 (GLP-1) has gained distinct recognition in reinstating glucose homeostasis in diabetes, while it is now largely accepted that it has an array of biological effects in the body. Several lines of evidence have revealed that GLP-1 and its analogs possess cardioprotective effects by various mechanisms related to cardiac contractility, myocardial glucose uptake, cardiac oxidative stress and ischemia/reperfusion injury, and mitochondrial homeostasis. Upon binding to GLP-1 receptor (GLP-1R), GLP-1 and its analogs exert their effects via adenylyl cyclase-mediated cAMP elevation and subsequent activation of cAMP-dependent protein kinase(s) which stimulates the insulin release in conjunction with enhanced Ca^2+^ and ATP levels. Recent findings have suggested additional downstream molecular pathways stirred by long-term exposure of GLP-1 analogs, which pave the way for the development of potential therapeutic molecules with longer lasting beneficial effects against diabetic cardiomyopathies. This review provides a comprehensive overview of the recent advances in the understanding of the GLP-1R-dependent and -independent actions of GLP-1 and its analogs in the protection against cardiomyopathies.

## 1. Introduction

Metabolic diseases have been on the rise globally, and novel therapeutics with lesser or no side effects to encounter the unmet medical needs of new-age morbid situations are highly needed [1,2]. From that perspective, growth factor signaling, particularly insulin signaling, is one of the most demanding pathways for cell growth, metabolism, and homeostasis. Nevertheless, any deviation or interruption of its normal processes could lead to the development of cardiometabolic disorders and premature death or senescence of the cells [3].

Glucagon-like peptide 1 (GLP-1) is an incretin peptide hormone that aids in insulin secretion, thereby contributing to the regulation and homeostasis of bioenergetic pathway and cell survival. In the human body, GLP-1 is mainly secreted by enteroendocrine L cells in the distal intestine, pancreatic α-cells, and the central nervous system (CNS) [4]. After a meal, GLP-1 is immediately released, leading to an increase in glucose-stimulated insulin secretion (GSIS) at physiological plasma levels. Previous studies suggested that factors from the gut provoke the pancreas to aid in nutrient assimilation, which laid the foundation for the treatment of diabetic patients using extracts from porcine small intestine [5]. The first incretin hormone, gastric inhibitory polypeptide (GIP), isolated from crude extracts of the porcine small intestine, inhibited gastric acid secretion [6]. Later, it was found that GIP stimulated insulin secretion in both animals and humans; it was renamed as glucose-dependent insulinotropic polypeptide; however, the acronym was remained as the same [7].

The second incretin hormone, GLP-1, was discovered in the pancreatic islets from anglerfish. GLP-1 and GLP-2 were identified in hamsters and humans; however, GLP-1 is the main isoform that could stimulate insulin secretion upon diet ingestion [8,9]. GLP-1 is produced in two bioactive forms consisting of glycine-extended GLP-1(7-37) and amidated GLP-1(7-36) via the post-translational processing of proglucagon by proprotein convertase subtilisin-kexin type 1 (PCSK1) or type 3 (PCSK3) [10,11] (Figure 1). GLP-1 receptor (GLP-1R) is classified in class B of the G protein-coupled receptor (GPCR) family that is abundant in the CNS and pancreas, while it is also found in lower expression in the heart, lungs, gut, muscles, kidneys, liver, peripheral nervous system (PNS), and other tissues [3]. GLP-1R gene expressions (transcripts per million, TPM) amongst females and males in the cardiovascular system and pancreas (from GTEx Portal https://gtexportal.org/home/gene/GLP1R accessed on 21 November 2022) are shown in Figure 2. Upon binding to GLP-1R, GLP-1 and its analogs initiate a series of events, including the activation of membrane-bound adenylyl cyclase (AC) and the subsequent production of cyclic adenosine monophosphate (cAMP). Several signal transduction pathways can be initiated downstream of cAMP formation via its effectors such as protein kinase A (PKA) and exchange protein activated by cAMP (Epac) [12,13].

GLP-1(7-36) has a relatively short half-life, which is cleaved by the activity of enzyme dipeptidyl peptidase-4 (DPP-4) at its N-terminus to produce GLP-1(9-36). It was previously supposed that GLP-1(9-36) is an inactive GLP-1 metabolite due to its poor interaction with GLP-1R and lack of significant effects on stimulating insulin secretion or glucose homeostasis [14]. However, recent studies showed that GLP-1(9-36) exerts cardioprotective effects against oxidative injury in H9c2 cardiomyoblasts through the phosphoinositide 3-kinase (PI3K)/Akt/nitric oxide synthase (NOS) pathway in an GLP-1R independent mechanism [15]. Similar cardioprotective roles of GLP-1(9-36) have been observed in dog models with dilated cardiomyopathy [16] and cultured cardiac cells of Goto–Kakizaki rats [17]. It has been suggested that GLP-1(9-36) amide is an active form at physiological concentration, while other GLP-1 metabolites, such as GLP-1(28-36) and GLP-1(32-36), are metabolically inactive [18]. However, growing evidence suggested that some of these metabolites may contribute to pleiotropic effects of GLP-1, irrespective of the GLP-1R [19,20]. At pharmacological doses, these metabolites possess glucoregulatory and cytoprotective effects such as reducing oxidative stress in vascular tissues [21], protecting pancreatic β-cells [22], and suppressing oxidative stress and gluconeogenesis in liver [23].

After a meal, GLP-1 is secreted from the L-cells in a biphasic pattern that is an early immediate rise of GLP-1′s plasma concentration starting approximately 10 to 15 min after ingestion, followed by a late increase achieving the maximum level at 30–60 min, and then a gradual decline to baseline over several hours [24]. However, due to local GLP-1 degradation by DPP-4 and further degradation by DPP-4 in the liver, only 10–15% of endogenously released GLP-1 reaches the systemic circulation [25]. Due to the rapid degradation of GLP-1 by DPP-4, exogenously administered GLP-1 has a short half-life of 1–2 min. Therefore, several GLP-1 analogs (or GLP-1 mimetics) have been developed in two ways, GLP-1R agonists resistant to DPP-4-mediated degradation and DPP-4 inhibitors, both of which tend to prolong the retention time of circulating GLP-1(7-36) amide [26]. To avoid degradation by DPP-4, GLP-1 analogs have been developed by modifying the amino acid residues in the native GLP-1. Additionally, they were synthetically created through the modifications of the structure of exendin-4, a 39-amino-acid peptide initially obtained from the saliva of the lizard *Heloderma suspectum*, which activated GLP-1R and is inherently resistant to DPP-4 degradation [27,28].

GLP-1 analogs such as liraglutide and exendin-4 are not rapidly degraded by DPP-4, allowing them to exert long therapeutic effects that would be impossible with endogenous GLP-1 due to their extremely short half-life in systemic circulation [29,30]. GLP-1 analogs exert a plethora of additional beneficial effects beyond antidiabetic effects, including increased satiety, decreased gastric emptying, replenishment of insulin stores, cytoprotective properties, and anti-inflammatory effects [31,32,33]. Currently, there are various GLP-1 analogs approved for the treatment of type 2 diabetes (T2DM) which were developed from human GLP-1 backbone (_glutide) (e.g., albiglutide, dulaglutide, liraglutide, and semaglutide) and exendin-4 backbone (_natide) (e.g., exenatide, efpeglenatide, and lixisenatide) (Table 1). These drugs are considered more efficient than DPP-4 inhibitors that block the breakdown of GLP-1; instead, they directly activate the GLP-1R signaling [34,35].

## 2. GLP-1 Analogs and Physiological Importance in Humans

GLP-1 and its analogs have emerged as attractive approaches to target the secondary manifestations due to hyperglycemic or diabetic conditions in the human body. Beyond their antidiabetic effects, GLP-1 analogs not only reduce diet-induced pancreatic secretions, but also diminish gastric emptying [40]. The ileal-brake activity of GLP-1 was confirmed using the GLP-1R antagonist, exendin-(9-39), which was suggested to have physiologic significance [41,42]. The inhibitory action of GLP-1 was later discovered to be mediated by a vagal pathway [43]. The systemic effects of GLP-1 and its analogs on various vital tissues and organs include reduction of insulin resistance, decrease in hyperglycemia, loss of body weight, reduction of blood pressure, decrease in reactive oxygen species (ROS) production, modulation of the inflammatory response, and improvement of reproductive function (Table 2, Figure 3) [38,44,45].

An endogenous GLP-1 is produced not only peripherally in the ileum, but also centrally in the brain, particularly in the area of nucleus of the solitary tract (NTS) of the hindbrain and the olfactory bulb, thereby controlling appetite and feeding behaviors [46]. The GLP-1 analog promotes satiety by activating GLP-1Rs in the hypothalamus and brainstem, resulting in a decline of food intake and weight loss [11,47]. Besides its metabolic functions in the brain, GLP-1 directly exhibits neuroprotective and neurotropic effects [48]. GLP-1 or exendin-4 administration into the lateral ventricles of mice reduced endogenous Aβ levels and prevented Aβ-induced cell death in rat hippocampus neurons [49]. During hypoxic injury, exendin-4 increased cell viability in cultured embryonic primary cerebral cortical and ventral mesencephalic (dopaminergic) neurons and protected against metabolic and oxidative damages. Importantly, these protective effects were not observed in neurons from GLP-1R knockout mice [50]. Exendin-4 also preserved cell viability and decreased apoptosis induced by hydrogen peroxide (H_2_O_2_)-induced oxidative stress in NSC19 neuroblastoma cells and in SOD1-G93A mutant mice models of amyotrophic lateral sclerosis [51,52].

The role of GLP-1 in the CNS, where it regulates satiety and body weight, is one of the intriguing aspects of its biology [47,53]. After activation of GLP-1Rs in the portal vein, this brain–gut axis regulates muscle glycogen synthesis and whole-body glucose clearance [54,55]. However, the relative importance of GLP-1Rs in the CNS and PNS in maintaining energy homeostasis is unknown. Previous studies have suggested that there is improved vascular function upon administration of GLP-1 in subjects with normal glucose tolerance and T2DM patients [56,57,58]. There was increased blood pressure and heart rate upon intermittent GLP-1 administration, while a modest increase in heart rate and a decrease in blood pressure was observed upon chronic activation of the GLP-1R with GLP-1 analogs [59]. This observation has raised the curiosity of many interested in the potential beneficial effects of GLP-1 on the heart. Evidently, GLP-1 and GLP-1Rs are abundantly distributed in many vital tissues, and their physiologic actions have gradually been elucidated, particularly since the advent of GLP-1 mimetics, including GLP-1R agonists and DPP-4 inhibitors.

GLP-1R expresses in various immune cells, including splenocytes, thymocytes, regulatory T cells, bone marrow-derived cells, macrophages, and invariant natural killer T cells, and thereby modulates the immune responses [60,61]. The GLP-1 analog, liraglutide, improved the psoriasis area and severity index and reduced the inflammatory cytokine secretion from natural killer T cells in patients with psoriasis [62]. In a mice model of obesity (high fat diet, HFD), liraglutide decreased cardiovascular risk factors including insulin resistance and inflammation, reduced monocyte vascular adhesion, and improved cardiac function by activating adenosine 5′-monophosphate activated protein kinase (AMPK) [63]. Moreover, liraglutide has been approved for chronic weight management in patients with obesity or weight related comorbid conditions including hyperlipidemia, hypertension, or T2DM [64,65]. Another GLP-1 analog, exndin-4, reduced mRNA levels of the proinflammatory cytokines, tumor necrosis factor-α (TNF-α), monocyte chemoattractant protein 1 (MCP-1), and signal transducer and activator of transcription 3 (STAT-3) in HFD fed mice [66]. Additionally, the number of transforming growth factor-β (TGF- β) positive cells were increased upon GLP-1/gastrin treatment in islet transplanted nonobese diabetic animals [67].

GLP-1 analogs stimulate natriuresis at physiological levels by inhibiting sodium reabsorption via decreased activity of sodium–hydrogen exchanger 3 (NHE3) or solute carrier family 9 member 3 (SLC9A3) in the proximal tubules, resulting in decreased blood pressure in patients with diabetes [68,69]. It is yet to be confirmed whether this effect is caused by direct GLP-1R activation in the kidneys. However, the involvement of renin–angiotensin system and a neural pathway has been suggested by previous studies [68,69].

## 3. Molecular Mechanisms Underlying Cardioprotective Effects of GLP-1 and Its Analogs

Cardiac muscle is predisposed to damage and dysfunction under several metabolic stresses, particularly under diabetic conditions and biological aging. Based on intense lines of preclinical studies both in vitro (Table 3) and in vivo (Table 4), GLP-1 analogs expressed their cardiovascular benefits via several cytoprotective molecular mechanisms.

### 3.1. GLP-1 and Its Analogs and Cardiac Functions

Clinical research has demonstrated that GLP-1R analogs improve the prognosis of cardiovascular disorders by raising the resting heart rate and decreasing arrhythmia in patients with heart failure (HF) [25,90]. Patients with HF who received GLP-1R analog for 5 weeks have shown improvements in their left ventricular ejection fraction and 6-min walking distance as well as increases in their maximal oxygen uptake [91]. Albiglutide administration for 12 weeks in patients with HF elicits an improvement in their cardiac oxygen consumption [92]. Exenatide lowered atherosclerotic levels and enhanced diastolic function in patients with T2DM [93]. Mechanistically, GLP-1 analogs improve the sympathetic nervous system, which, in turn, causes an increase in heart rate [94,95]. While the systemic effects and increased heart rate elicited by exendin-4 treatment in vivo were eliminated by blocking β-adrenergic receptor (β-AR) [96], GLP-1 treatment via peripheral or intracerebroventricular routes dose-dependently increased blood pressure and heart rate in animal models [95,97,98]. The cause of an elevated heart rate has remained intriguing; however, it might be related to the stimulation of sympathetic nervous system. GLP-1 analogs (such as exendin-4) activated c-fos expression in the adrenal medulla and medullary catecholamine neurons, indicating that central GLP-1Rs mediate neuroendocrine responses and influence sympathetic outflow [95].

Cardiovascular risk factors such as dyslipidemia and hypertension have been demonstrated to be reduced by liraglutide and exenatide. Exenatide revealed a drop in systolic blood pressure (SBP) without changes in diastolic blood pressure (DBP) in a meta-analysis of six trials including over 2000 participants, while weight loss could not entirely explain this effect [99]. However, another clinical study demonstrated that liraglutide expresses a decrease in SBP in the first two weeks of treatment independent of weight loss [100]. Exenatide has been shown to lower triglyceride levels by 12%, total cholesterol by 5%, low-density lipoprotein cholesterol by 6%, and increase high-density lipoprotein cholesterol by 24% in three-year trials, thus contributing to lowering blood pressure [101].

### 3.2. GLP-1 and Its Analogs and Myocardial Glucose Uptake

GLP-1 has been suggested to have a wide range of biological effects on several target organs, including heart, liver, brain, muscles, and adipose tissues [44,102,103]. The classic response elicited by GLP-1R activation is the improvement of cardiac function by increasing glucose uptake and coronary flow as well as the secretion of the main blood pressure and electrolyte regulator, atrial natriuretic peptide (ANP), in mice [69,104,105]. GLP-1-(7-36) amide was found to directly augment myocardial glucose uptake by increasing the translocation of glucose transporters (GLUTs), GLUT-1 and GLUT-4, in normal and ischemia/reperfusion (I/R)-induced ischemic rat hearts in a similar pattern as insulin [106]. Treatment with exendin-4 induced cardiac glucose uptake and ATP production via an increase in GLUT-1 translocation which was dependent on the expression of p38 mitogen-activated protein kinase (p38MAPK) in hypoxia/reoxygenation (H/R)-induced injury in H9c2 cardiomyoblasts [107]. However, clinical evidence suggests that baseline functions of glucose transport and insulin resistance are the main determinants of the impact of GLP-1 analog on glucose transport and glucose uptake in cardiomyocytes during hyperglycemic states in patients with T2DM [108] and during normo- or hypoglycemia in healthy subjects [109]. The occurrence of apoptosis in cardiac cells contributes to the development of diabetic cardiomyopathy, which is caused by the buildup of intracellular lipids such as saturated palmitic acids. In addition, treatment with GLP-1 analogs provoked β-catenin, signaling leading to an activation of the protein kinase B-glycogen synthase kinase-3β survival pathway and a decrease in apoptotic activation in neonatal rat cardiomyocytes exposed to palmitate [110].

### 3.3. GLP-1 Analogs and Cardiac Oxidative Stress as Well as Ischemia/Reperfusion (I/R) Injury

Previous studies have shown that GLP-1 and its analogs express cardiac-protective effects and are crucial in the regulation of heart functions. Exendin-4, for instance, reduced infarct size and enhanced mechanical function in isolated rat hearts exposed to I/R injury [111]. Likewise, exenatide infusion decreased the size of the infarct area found in a porcine model of I/R injury [112]. In addition, ejection fraction and other functional indicators were improved in patients with acute myocardial infarction (MI) after a 72-h GLP-1 infusion [113]. Moreover, treatment with GLP-1 analogs prevented the cardiomyocyte damage and apoptosis from H/R injury [114]. Liraglutide inhibited plasminogen activator inhibitor type-1 (PAI-1) and vascular adhesion molecules in vitro and in vivo and enhanced NOS activity [1]. Moreover, GLP-1R activation restored endoplasmic reticulum (ER) homoeostasis, cytoprotection, and the signaling pathways disrupted by various stress stimuli [63,115,116].

Oxidative stress ensuing from an imbalance between antioxidant defenses and the generation of ROS could result in cellular damage and apoptosis. The toxic levels of ROS are generated in patients of HF [117] and animal models of MI [118]. Recent studies have shown that GLP-1 analogs exert cardioprotective benefits through regulating the antioxidant and anti-apoptotic responses. Activation of the GLP-1R increased cAMP levels, thereby activating PKA and Epac-dependent antioxidant and anti-apoptotic cytoprotective responses against H_2_O_2_-induced oxidative damage in cardiomyocytes [71]. Although it has been reported that excessive stimulation of β-ARs results in the induction of insulin resistance in many cell types, including cardiomyocytes [119,120], adipocytes [121], and skeletal muscle [122], prolonged overstimulation of GLP-1R has not been shown to cause these detrimental effects.

GLP-1 appears to have both GLP-1R-dependent and -independent mechanisms for mediating its cardioprotective and vasodilatory effects [123]. The findings that GLP-1(9-36) maintained its vasodilatory and cardioprotective properties in hearts isolated from GLP-1R−/− mice serve as an explanation of this phenomenon [123]. In fact, GLP-1(9-36) enhanced left ventricular function and boosted myocardial glucose absorption in dogs with dilated cardiomyopathy [16]. In contrast to intact GLP-1, GLP-1(9-36) directly inhibited the generation of superoxide in human artery endothelial cells induced by high glucose or free fatty acids [124] and cytoprotective actions on mouse cardiomyocytes exposed to H_2_O_2_ [115]. This is consistent with the findings that GLP-1(9-36) leads to significant enhancement of the rate-pressure product when the GLP-1R antagonist, exendin (9-39), is administered, although the GLP-1R agonist exendin-4, which is DPP4 resistant, is no longer effective in increasing left ventricular end-diastolic pressure in isolated rat hearts [111], indicating the presence of a GLP-1R independent pathway. This is further supported by our recent findings, wherein exposure to GLP-1(9-36) inhibited H_2_O_2_-induced oxidative stress and apoptosis in H9c2 cardiomyoblasts via PI3K/Akt/NOS signaling pathway; therefore, GLP-1R analog treatment represents a potential therapeutic approach for the prevention of pathological oxidative stress and damage to the heart [15].

## 4. Significant Roles of GLP-1 and Its Analogs in Mitochondrial Homeostasis

Mitochondria is the energy generator for the cell, oxidizing various carbon sources and producing energy in the form of ATP. Mitochondria produces intermediate substrates to be utilized in anabolic or catabolic processes and regulates crucial physiological and pathological processes such as calcium homeostasis, metabolism of biomolecules (amino acids, lipids, and glucose), oxidative stress, apoptosis, aging, and autophagy [125]. The majority of the energy for cardiac functions is derived from the oxidation of fatty acids in mitochondria. However, in T2DM, insulin resistance leads to lower glucose utilization and oxidative phosphorylation; consequently, cardiomyocytes become reliant on the oxidation of fatty acids for energy supply. This shift in substrate utilization leads to an imbalance in the uptake and oxidation of fatty acids, which ultimately causes mitochondrial dysfunction. Moreover, chronic hyperglycemia instigates mitochondrial oxidative stress and fragmentation, leading to cellular dysfunction and apoptosis [2]. In diabetic hearts, the damaged mitochondria in cardiomyocytes resulted in the generation of excessive amounts of ROS causing pathological oxidative stress, which deteriorated the myocardial functions [126].

Therefore, repair of the damaged mitochondria and replenishment of new mitochondria would be an effective strategy to maintain a healthy pool of mitochondria to ameliorate the pathological outcomes of cardiac diseases. Our recent study showed that stimulation of GLP-1R using exendin-4 exhibits antioxidant and antiapoptotic effects, as well as an improvement in mitochondrial functions via cAMP/Epac/PI3K/Akt signaling in H9c2 cardiomyoblasts exposed with methylglyoxal, an advanced glycation end product (AGE). Since intracellularly accumulation of AGEs causes mitochondrial impairment, oxidative stress, and apoptosis, attenuating methylglyoxal-induced mitochondrial abnormalities in the heart by exendin-4 may help to ameliorate and delay the progression of heart diseases [70]. Several clinical studies have also supported that GLP-1 analogs are beneficial in lowering the occurrence of adverse cardiovascular events [127,128]. GLP-1 analogs with longer half-life, such as exenatide and liraglutide, have been designed to tolerate DPP-4 degradation and are efficacious in treating T2DM [129,130]. However, long-term usage is linked to an increased risk of hypoglycemia, pancreatic cancer, thyroid cancer, and gastric ailments [131,132].

Although the precise molecular mechanisms responsible for the cardioprotective benefits of GLP-1 analogs have not been clearly identified so far, there are preclinical studies (Table 3 and Table 4) which suggest that this drug class may also be effective when administered shortly after MI through the activation of the mitophagy process and the resurrection of non-performing mitochondria. In H/R model of H9c2 cells, pretreatment with exenatide reduced mitochondrial abnormalities by diminishing oxidative stress and mitochondrial calcium overload while increasing mitochondrial ATP synthase activity or ATP synthesis and rescuing mitochondrial membrane potential (MMP) [133]. Moreover, mitochondrial uncoupling protein-3 (UCP-3) and nuclear respiratory factor-1 (NRF-1) were increased while lactate dehydrogenase (LDH), creatine kinase-MB (CK-MB), and cardiomyocyte apoptosis were decreased [133]. Liraglutide has been reported to protect cardiomyocytes from interleukin-1β (IL-1β)-induced metabolic disturbance and mitochondrial dysfunction by enhancing phosphorylated AMPK and acetyl-coA carboxylase (ACC) as well as improving peroxisome proliferator-activated receptor-gamma coactivator-1α (PGC-1α), carnitine palmitoyltransferase-1 (CPT1), and diacylglycerol acyltransferase 1 (DGAT1) [134]. GLP-1 analog also decreased the expression of H/R-induced ER stress proteins, including glucose regulatory protein 78 (GRP78), CCAAT-enhancer-binding protein homologous protein (CHOP), and caspase-12, and exhibited cardioprotective effects by an activation of the GLP-1R/PI3K/Akt signaling pathway [114].

Hyperglycemic condition in primary cardiomyocytes has been shown to cause ER stress, mitochondrial oxidative stress, and enhanced mitochondria-associated ER membrane (MAMs)-associated proteins, including GRP75 and mitofusin-2 (MFN2). MFN2 played an important role in high glucose-induced ER stress in cardiomyocytes, while silencing of MFN2 abated mitochondrial calcium overload-mediated mitochondrial impairment, thereby decreasing ER stress-mediated cardiomyocyte apoptosis [135]. Furthermore, metabolic signaling pathways mediated by insulin-like growth factor 1 (IGF-1), mammalian target of rapamycin (mTOR), AMPK, and sirtuins are linked to the balance between nutrient anabolism and catabolism in cells, which maintain cellular homeostasis [136,137]. In both in vivo and in vitro studies, employing cardiac hypertensive model elucidated that administration of exendin-4 attenuated cardiac hypertrophy via the inhibition of mitochondrial permeability transition pore (MPTP) opening and ROS release [138].

Exendin-4 also was found to inhibit cardiac hypertrophy by upregulating GLP-1R expression and activating the AMPK signaling pathway [72]. Liraglutide could protect the heart against aortic banding-induced myocardial fibrosis and dysfunction by suppressing mTOR/p70S6K signaling and boosting autophagy [89]. Another study reported that liraglutide reduces cardiac hypertrophy via the angiotensin II/AT_1_R/ACE2 and AMPK/mTOR/ p70S6K pathways [76]. A recent study examined the efficacy of a newly synthesized GLP-1R agonist, compound DMB, in attenuating pathological left ventricular remodeling in a permanent coronary artery ligation mice model and suggested that increasing mitochondrial biogenesis and Parkin-associated mitophagy after MI might be a viable target for the therapeutic management of adverse cardiac remodeling [139]. In line with previous studies, liraglutide ameliorated cardiac fibrosis and cardiomyocyte death in the infarcted heart in rat, while their mechanistic findings in H9c2 cardiomyoblasts showed that SIRT1/Parkin/mitophagy pathways were involved in its cytoprotective effects [140]. Contrastingly, another study did not find any effects on adverse remodeling or cardiac function upon a similar liraglutide treatment in rats [140]. Nevertheless, excessive glucose exposure caused increased intracellular ROS, apoptosis, MMP loss, decreased autophagy, and increased phospho-mTOR (p-mTOR) and p-ULK1 levels, all of which were counteracted by GLP-1 analogs. These findings suggested that GLP-1 analogs could reverse glucose toxicity by inducing mTOR/ULK1-dependent autophagy [141].

## 5. Co-Agonists: A Single Molecule Stimulating Multiple Peptide Receptors

There have been propositions that multiple signaling pathways would need to be targeted for an efficient clinical treatment of obesity due to the system’s redundancy in regulating food intake and body weight [142]. Using this concept, the most delicate way to treat T2DM was to combine activity of GLP-1 with that of the GIP to achieve long-lasting beneficial effects. A synthetic peptide with stimulatory activity at both the GIP receptor and GLP-1R proved to be more effective at reducing body weight and blood glucose in obese mice compared to a selective GLP-1R agonist [143]. Additionally, recent studies in mice have demonstrated that GIP receptor is expressed in hypothalamic neurons, and that activating it significantly reduced food intake and body weight, when combined with GLP-1 [144,145,146]. These initial observations, together with subsequent studies, led to the development of multi-receptor peptides affecting more than one intracellular signaling pathway, which have become the choice of treatments for T2DM and obesity.

Following the initial studies in multi-receptor peptides, a number of single-molecule GIP receptors and GLP-1R agonists were created. Tirzepatide, a dual GIP/GLP-1 receptor co-agonist for the treatment of T2DM, is a 39-amino acid linear peptide that is comparable in size to the related hormones GIP and GLP-1 [147]. In murine pancreatic islets from GIP receptor knockout mice, GLP-1R antagonist inhibited the insulin secretion stimulated by tirzepatide, while in islets lacking GLP-1Rs, GIP receptor blocker suppressed the insulin secretion upon tirzepatide stimulation. Moreover, tirzepatide decreased both food intake and body weight more prominently than selective GLP-1R agonists such as semaglutide in HFD mice [51]. Tirzepatide stimulated cAMP production in pancreatic ß-cells, similar to a combination of GLP-1 and GIP [51], while also eliciting better efficacy in controlling cardiovascular disease risk factors such as glucose levels and body weight loss in diabetic patients [148,149].

Tirzepatide ameliorated a number of cardiovascular risk factors over selective GLP-1R agonists. In blood samples analyzed from diabetic patients, triglycerides, apolipoprotein C-III, phosphatidylethanolamines/phosphatidylcholines, circulating levels of branched-chain amino acids (leucine, isoleucine, and valine), inflammatory markers (e.g., C-reactive protein), and adhesion molecules (e.g., intercellular adhesion molecule-1; ICAM-1) were reduced by tirzepatide [150,151,152]. Additionally, a preliminary proof of tirzepatide’s cardiovascular safety was elucidated by a recent meta-analysis combining all the clinical trials. The commonly specified cardiovascular end points were projected to have a risk below 1.0, and for 4-point major adverse cardiovascular events (MACE) (MI, stroke, hospitalization for angina, and all-cause death) and 3-point MACE (MI, stroke, and all-cause mortality), the hazard ratios were less than 1.3 [153].

## 6. Conclusions and Future Directions

Inadequate exercise, high fat and highly processed diet, and excess nutrition all contribute to the rising prevalence of T2DM, which severely reduces both life expectancy and quality of life. T2DM is linked to a large worldwide mortality burden, which primarily involves cardiovascular events including HF and stroke. Therefore, new therapeutic modalities are highly required. GLP-1 analogs are relatively novel medications used for the treatment of T2DM. We summarized the recent findings pertaining to cardioprotective effects as well as molecular mechanisms of GLP-1R-dependent and -independent pathways (Figure 4). GLP-1 analogs are able to directly affect not only other important cardiovascular risk factors, including high blood pressure, dyslipidemia, or obesity, but also the hyperglycemia in diabetic patients. These characteristics were exemplified in several cardiovascular outcome trials for exendin-4, liraglutide, semaglutide, dulaglutide, albiglutide, and tirzepatide, all of which demonstrated the improvement of cardiovascular prognosis in addition to safety. Although the effects of the GLP-1R agonists were well understood, further research is necessary to fully understand their GLP-1R-dependent and -independent effects on the cardiovascular system, particularly with regard to their signaling pathways. Recently, GLP-1R-independent pathways have been identified to have direct vasodilatory and cardioprotective effects, as well as anti-inflammatory and antioxidant benefits. The protective properties of GLP-1 analogs can be thoroughly investigated using contemporary omics technologies to reveal epigenetic, metabolomics, and proteomic changes to explain the observed beneficial effects. The discovery of additional GLP-1 signaling pathways (unrelated to glucose lowering effects) is very likely to make this class of medication relevant for the treatment of several chronic diseases, including obesity, T2DM, cardiovascular diseases, cancer, neurodegenerative and neuroinflammatory diseases, and others.

## Figures and Tables

**Figure 1 pharmaceuticals-16-00836-f001:**
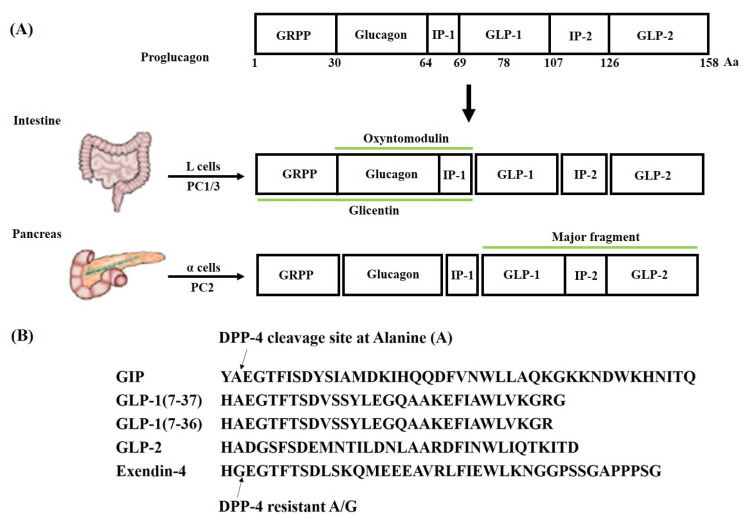
**Structural arrangement of mammalian proglucagon.** (**A**) The amino acid sequences in proglucagon have been marked as numbers. The post-translational processing produces peptides in the intestine and pancreas as shown. (**B**) The DPP-4 cleavage site in the amino acid chain is indicated by arrows. GRPP, glicentin-related pancreatic polypeptide; IP, intervening peptide; PC1/2/3, prohormone convertases 1/2/3; DPP-4, dipeptidyl peptidase-4.

**Figure 2 pharmaceuticals-16-00836-f002:**
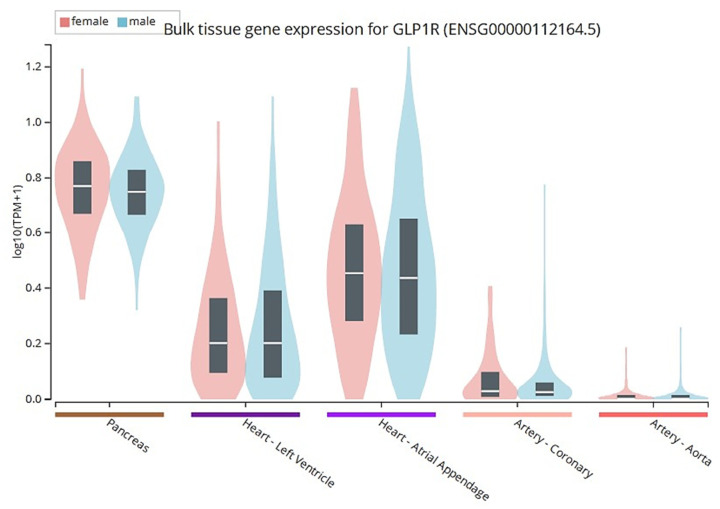
GLP-1 receptor (GLP-1R) gene expression (transcripts per million; TPM) amongst females and males in selected tissues related to the cardiovascular system and pancreas (from GTEx Portal https://gtexportal.org/home/gene/GLP-1R accessed on 21 November 2022). Box plots are depicted as median and correspond to 25th and 75th percentiles. The pancreas expresses the higher expression (median value, 4.5) when compared to heart-atria (median value, 1.7), heart left ventricle (median value, 0.6), coronary artery (median value, 0.05), and aorta (median value, not available). In this cohort analysis, there are no significant sex differences in the expression pattern of GLP-1R amongst males and females.

**Figure 3 pharmaceuticals-16-00836-f003:**
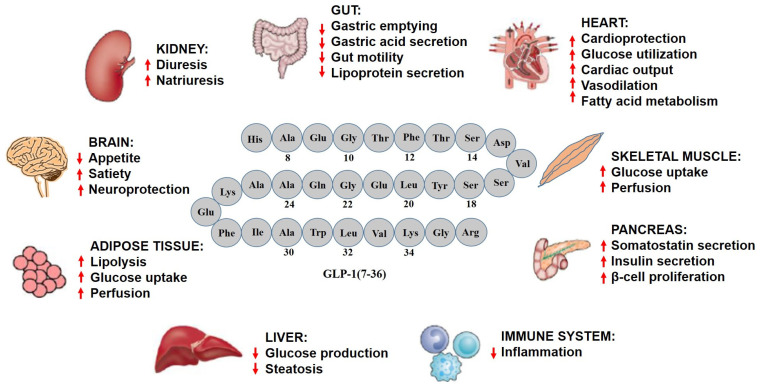
**Systemic effects of GLP-1 and its analogs.** The effects on various tissues/organs illustrated in the figure are based on results obtained in several preclinical or clinical studies at physiological GLP-1 levels or after exposure to GLP-1R analogs. GLP-1R is abundantly distributed in the tissues/organs of the peripheral and central nervous system with the different expression levels leading to the distinct biological functions in each tissue/organ. In the CNS and pancreas, high expression of GLP-1R is detected, while lower expression of GLP-1R is observed in the heart, lungs, gut, muscle, kidneys, liver, PNS, and other tissues. ↑, Increase; ↓, Decrease.

**Figure 4 pharmaceuticals-16-00836-f004:**
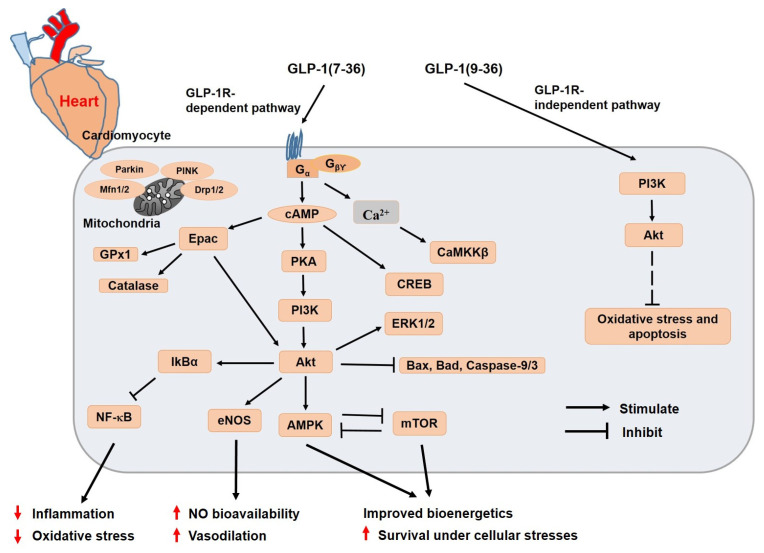
**Schematic diagram representing cardioprotective molecular mechanisms of GLP-1 analogs in cardiomyocytes.** The GLP-1R-dependent and -independent signaling pathways have been described as contributing to healthy myocardium and heart functions under pathophysiological conditions. The binding of GLP-1 to its receptor activates adenylate cyclase, and cAMP levels are elevated, which is followed by the activation of downstream cAMP sensitive molecular effectors in a GLP-1R-dependent way. While GLP-1R-independent GLP-1(9-36) activates PI3K, Akt, and the downstream signaling pathway, the full mechanisms of this process are not yet known. GLP-1, glucagon-like peptide-1; cAMP, cyclic adenosine monophosphate; Akt, protein kinase B; Epac, exchange protein activated by cAMP; CREB, cAMP-response element binding protein; CaMKKβ, calcium ions/calmodulins dependent protein kinase kinases β; PINK1, PTEN-induced kinase 1; Mfn, Mitofusin; Drp, dynamin-related protein; NF-κB, nuclear factor-κB; eNOS, endothelial nitric oxide synthase; AMPK, AMP-activated protein kinase; mTOR, mammalian target of rapamycin; ERK, extracellular-signal-regulated kinase; PI3K, phosphatidylinositide 3-kinase.

**Table 1 pharmaceuticals-16-00836-t001:** List of current GLP-1 and its analogs [36,37,38,39].

GLP-1 Analogs	Company	FDA Approval Year for T2DM	Description	Administration	Half-Life
**Human GLP-1 backbone (_glutide)**					
Albiglutide	GlaxoSmithKline	2014	Extended release, fused to human albumin	Subcutaneous (SC), once weekly	5–6 days
Dulaglutide	Eli Lilly	2014	Extended release, Fc region of human IgG4	SC, once weekly	4.7 days
Liraglutide	Novo Nordisk	2010	Immediate release, linked with free fatty acid	SC, once daily	12–13 h
Semaglutide	Novo Nordisk	2017	Extended release, linked with free fatty acid	SC, once weekly	5–7 days
	Novo Nordisk	2020	Extended release, linked with free fatty acid	Oral, once daily	5–7 days
**Exendin-4 backbone (_natide)**					
Exenatide	Eli Lilly	2005	Immediate release	SC, twice daily	2.4 h
	AstraZeneca	2012	Extended release, encapsulated in microsphere form	SC, once weekly	3–5 days
Efpeglenatide	Sanofi	Not available	Extended release	SC, once monthly	Not available
Lixisenatide	Sanofi	2016	Immediate release, linked with poly-lysine tail	SC, once daily	3–4 h

**Table 2 pharmaceuticals-16-00836-t002:** Effects of GLP-1 and its analogs on organs/tissues [38,44,45].

Organs/Tissues	Biological Effects
Heart	▪ Protect against cardiac ischemia and myocardial damage; ▪ Enhance heart rate and cardiac output; ▪ Improve glucose utilization and increase fatty acid metabolism; ▪ Vasodilation and anti-inflammatory effect; ▪ Increase atrial natriuretic peptide (ANP) secretion.
Brain	▪ Suppress appetite and increase satiety; ▪ Neuroprotection, improve learning and memory; ▪ Induce proliferation of neural stem cells; ▪ Decrease inflammation.
Liver	▪ Decrease glucose production; ▪ Reduce liver fat and steatosis.
Kidneys	▪ Diuresis and natriuresis.
Pancreas	▪ Enhance somatostatin and insulin secretion; ▪ Inhibit apoptosis and increase proliferation of β-cells.
Gut	▪ Delay gastric emptying time and acid secretion; ▪ Inhibit peristalsis and lipoprotein secretion.
Adipose tissues	▪ Increase lipolysis and glucose uptake.
Immune system	▪ Reduce inflammation.
Skeletal muscle	▪ Induce glucose uptake and tissue perfusion.
Reproductive system	▪ Improve ovarian function; ▪ Increase menstrual frequency and ovulation rates; ▪ Decrease infertility.

**Table 3 pharmaceuticals-16-00836-t003:** Preclinical studies of GLP-1R agonists for cardioprotective effects (*In vitro*).

GLP-1 Analogs	Study Models	Mechanistic Findings	Ref.
Exendin-4	Methylglyoxal-exposed H9c2 cardiomyoblasts	▪ Inhibits intracellular and mitochondrial ROS production, apoptosis, and MMP impairment; ▪ Upregulates genes related to mitochondrial functions and dynamics via cAMP/Epac/PI3K/Akt pathway.	[70]
Exendin-4	Hydrogen peroxide (H_2_O_2_) -exposed neonatal rat cardiomyocytes	▪ Exhibits antioxidant effects by inducing the synthesis of catalase, GPx-1, and Mn-SOD via Epac signaling; ▪ Elicits anti-apoptotic effects by enhancing Bcl-2 expression and inhibiting caspase-3 activity via PKA and Epac signaling.	[71]
Exendin-4	Phenylephrine-induced neonatal rat cardiomyocyte	▪ Attenuates cardiac hypertrophy by upregulating GLP-1R expression and activating AMPK/mTOR signaling.	[72]
Exendin-4	Oxygen glucose deprivation/reoxygenation (OGD/R) in human ventricular cardiomyocytes	▪ Restores autophagic flux induced by OGD/R through promoting nuclear translocation of TFEB; ▪ Reduces infarct size and preserves cardiac function through anti-apoptosis and antioxidative effects.	[73]
Exendin-4	High-density lipoprotein (HDL)-induced human umbilical vein endothelial cells	▪ Protects endothelial function by increasing HDL scavenger receptor class BI expression via AMPK/FoxO1 pathway and activating eNOS.	[74]
GLP-1	HL-1 cells derived from mouse atrial cardiac muscle cells	▪ Regulates arrhythmogenesis through modulating calcium handling proteins.	[75]
Liraglutide	Angiotensin II-treated H9c2 cells	▪ Decreases cardiac hypertrophy by increasing GLP-1R expression and activating AMPK; ▪ Reduces the expression of AngII/AT_1_R/ACE2 and mTOR.	[76]
Semaglutide	Lipopolysaccharides-treated H9c2 cells	▪ Protects exercise-induced cardiomyopathy by activating AMPK pathway; ▪ Increases autophagy, reduces ROS production and inflammatory markers (NF-κB, TNF-α, and IL-1β).	[77]
Semaglutide	Hypoxia/reoxygenation (H/R) injury in H9c2 cells	▪ Inhibits H/R injury-induced cardiomyocyte apoptosis by activating PKG/PKCε/ERK1/2 pathway.	[78]

**Table 4 pharmaceuticals-16-00836-t004:** Preclinical studies of GLP-1R agonists for cardioprotective effects (*In vivo*).

GLP-1 Analogs	Study Models	Mechanistic Findings	Ref.
Exendin-4	Left anterior descending (LAD) coronary artery ligation-induced heart failure rats	▪ Attenuates cardiac remodeling through activation of eNOS/cGMP/PKG pathway.	[79]
Exendin-4	LAD coronary artery ligation-induced myocardial infarction (MI) rats	▪ Inhibits collagen I, collagen III, transforming growth factor-β1, phosphorylated PI3K, and Akt levels.	[80]
Exendin-4	High-fat diet (HFD)-induced diabetic rats	▪ Inhibits cardiomyocyte pyroptosis via AMPK-TXNIP pathway.	[81]
Exendin-4	LAD coronary artery ligation-induced MI rats	▪ Attenuates myocardial remodeling after an acute MI by activating β-arrestin-2, protein phosphatase 2A, and GSK-3, and inhibiting β-catenin.	[82]
Exendin-4	Streptozotocin (STZ)-induced diabetic rats	▪ Downregulates NF-kB and upregulates cardiolipin, PGC1α, LC3, and beclin.	[83]
Exendin-4	LAD coronary artery ligation-induced MI rats	▪ Reduces ventricular arrhythmias through decreasing SR calcium leak, reduced RyR2 phosphorylation, and decreased CaMK-II activity.	[84]
Exenatide	DPP4-deficient rats; Adiponectin-deficient mice	▪ Prevents chronic stress induced vascular senescence.	[85]
Exendin-4	LAD coronary artery ligation-induced MI rats	▪ Decreases susceptibility to atrial arrhythmogenesis; ▪ Improves conduction properties and exerts antifibrotic effects via GLP-1R signaling.	[80]
Exendin-4	LAD coronary artery ligation-induced MI rats	▪ Increases GLP-1/GLP-1R expression in ventricular tissues; ▪ Activates eNOS/cGMP/PKG signaling and inhibits CaMKII pathway; ▪ Inhibits pathological cardiac remodeling;	[79]
Liraglutide	Spontaneously hypertensive rats (SHR)	▪ Delays progression of hypertension through activation of brainstem dopamine beta-hydroxylase (DBH) neurons; ▪ Suppresses sympathetic nerve activity.	[86]
Liraglutide	Apolipoprotein E deficient (ApoE−/−) mice	▪ Suppresses atherosclerotic lesions; ▪ Increases AMPK phosphorylation in aortic wall.	[87]
Liraglutide	HFD-feeding and STZ-induced diabetic rats	▪ Reduces myocardial damage by activating the Sirt1/AMPK pathways; ▪ Inhibits cellular pyroptosis.	[88]
Liraglutide	Abdominal aortic constriction-induced cardiac fibrosis rats	▪ Inhibits cardiac fibrosis and dysfunction by inhibiting mTOR/p70S6K signaling and enhancing autophagy.	[89]

## Data Availability

All data generated or analyzed during the current study are included in this published article.

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
