# Peer review of "Multifaceted Roles of GLP-1 and Its Analogs: A Review on Molecular Mechanisms with a Cardiotherapeutic Perspective"

_pharmaceuticals, 2023, doi:10.3390/ph16060836_

Round 1
Reviewer 1 Report
Minor English editing is required
Line 211 triglycerides level
Minor English editing is required
Author Response
Reviewer 1
Minor English editing is required.
Line 211 triglycerides level. Revised
We would like to thank the reviewer for reviewing our manuscript and raising these concerns. We have carefully revised the manuscript and also asked a native English speaker to check the language use and edit our manuscript in accordance with the reviewer’s suggestions. We hope the reviewer will acknowledge our effort to improve the usage of the English language in this manuscript.

Reviewer 2 Report
The authors have presented a well-written article with complementary figures and tables. The article provides a good summary of GLP-1 analogues with adequate emphasis and information on cardiovascular health.
1. Authors can improve how the article reads by correcting few grammatical errors, for example, using present instead of past tense when referring to figures in the manuscript.
2. More recent effects of GLP-1 also includes its role in fertility and reproductive function. It will be good to add some information either in text or in table.
3. Since the article is talking about multifaceted roles of GLP-1, there can be mention of where its receptors are present throughout the body with a table or figure.
4. Authors state that ‘Within couple minutes of ingesting nutrients, the plasma concentration of GLP-1 rises’. There is no mention of this in the article cited and usually GLP-1 rises within 15 mins after meal ingestion. Hence, it would be better if this was rephrased with a mention of when concentrations peak after a meal.
5. Figure 3 can also include current information on its reproductive effects, example, administration of GLP-1R agonists in women with PCOS.
6. The ileal brake activity of GLP-1 mentioned in line 135 might read better after line 123 when talking about its effects on gastric emptying as this mechanism is essential in reducing gastric emptying. It doesn’t fit well when discussing activity in the brainstem and hypothalamus unless an associated neural aspect is mentioned.
7. Authors can also mention the primary source of endogenous GLP-1 in the brain that will help better understand effects of GLP-1R agonists in the brain.
8. Paragraph starting with Line 164 can precede the paragraph starting with Line 148 to avoid discontinuity as immune cells incidentally come into the picture before the authors go back to the CNS.
9. in ‘GLP-1 analogs and cardiac glucose uptake’, the authors can add more information on mycocardial glucose uptake, if it increases only in patients with decreased glucose uptake and touch upon the role of glucose transporters in mediating this effect in the heart.
10. Line 219 seems quite repetitive as wide range of GLP-1 effects have already been mentioned in the article previously.
11. The authors can also mention the cardiovascular risk factors associated with GLP-1R agonists and how it might affect diabetic patients with pronounced cardiovascular issues.
Authors can improve how the article reads by correcting few grammatical errors, for example, using present instead of past tense when referring to figures in the manuscript.
Author Response
Reviewer 2
The authors have presented a well-written article with complementary figures and tables. The article provides a good summary of GLP-1 analogues with adequate emphasis and information on cardiovascular health.
We thank the reviewer for the insightful comments. We have addressed the concerns raised by the reviewer by adding explanations to each query. Our point-by-point response is presented below, and we have highlighted (marked in yellow) where modifications to the manuscript text have been made.
- Authors can improve how the article reads by correcting few grammatical errors, for example, using present instead of past tense when referring to figures in the manuscript.
We thank the reviewer for raising these concerns. We have carefully revised the manuscript and also asked a native English speaker to check the language use and edit our manuscript in accordance with the reviewer’s suggestions. We hope the reviewer will acknowledge our effort to improve the usage of the English language in this manuscript.
- More recent effects of GLP-1 also includes its role in fertility and reproductive function. It will be good to add some information either in text or in table.
We have agreed with reviewer’s constructive suggestion. We have included the impact of GLP-1 on the reproductive system in the text and Table 2 as follows.
Text (2. GLP-1 analogs and physiological importance in humans):
“The systemic effects of GLP-1 and its analogs on various vital tissues and organs include reduction of insulin resistance, decrease in hyperglycemia, loss of body weight, reduction of blood pressure, decrease in reactive oxygen species (ROS) production, and modulation of the inflammatory response, and improvement of reproductive function (Table 2) [Papaetis et al., 2022].” (Page 5, Line 137)
Table 2 (Effects of GLP-1 and its analogs on organs/tissues) (Page 5, Line 139)
Organs/tissues: Reproductive system
Biological effects: Improve ovarian function
Increase menstrual frequency and ovulation rates
Decrease infertility
Reference:
- Papaetis GS, Kyriacou A. GLP-1 receptor agonists, polycystic ovary syndrome and reproductive dysfunction: Current research and future horizons. Adv Clin Exp Med. 2022;31(11):1265-1274.
- Since the article is talking about multifaceted roles of GLP-1, there can be mention of where its receptors are present throughout the body with a table or figure.
Since GLP-1R is widely distributed in various tissues throughout the body, including the lungs, kidneys, central nervous system, cardiovascular system, gastrointestinal tract, and skin and vagus nerves, where it expresses the distinct biological functions [Campbell et al., 2013]. We have revised the figure caption of the Figure.3 in order to highlight the distribution breadth of GLP-1R which leads to the diversity and importance of its biological functions in each tissue/organ as follows.
Figure 3 (Figure caption):
“GLP-1R is abundantly distributed in the tissues/organs of the peripheral and central nervous system with the different expression levels leading to the distinct biological functions in each tissue/organ. In the CNS and pancreas, high expression of GLP-1R is detected, while lower expression of GLP-1R is observed in the heart, lungs, gut, muscle, kidneys, liver, peripheral nervous system, and other tissues.” (Page 6, Line 144)
Reference:
- Campbell JE, Drucker DJ. Pharmacology, physiology, and mechanisms of incretin hormone action. Cell Metab. 2013;17(6):819-837.
- Authors state that ‘Within couple minutes of ingesting nutrients, the plasma concentration of GLP-1 rises’. There is no mention of this in the article cited and usually GLP-1 rises within 15 mins after meal ingestion. Hence, it would be better if this was rephrased with a mention of when concentrations peak after a meal.
Thanks for highlighting these issues. We have changed to the relevant reference and revised the text according to the reviewer’s suggestion as follows.
Text (1. Introduction):
“After meal, GLP-1 is secreted from the L-cells in a biphasic pattern that is an early immediate rise of GLP-1’s plasma concentration starting approximately 10 to 15 minutes after ingestion, followed by a late increase achieving the maximum level at 30-60 minutes, and then a gradually decline to baseline over several hours [Nauck et al., 2011].” (Page 3, Line 95)
Reference:
- Nauck MA, Vardarli I, Deacon CF, Holst JJ, Meier JJ. Secretion of glucagon-like peptide-1 (GLP-1) in type 2 diabetes: what is up, what is down? Diabetologia. 2011;54(1):10-8.
- Figure 3 can also include current information on its reproductive effects, example, administration of GLP-1R agonists in women with PCOS.
Due to the space limitation of Figure.3, we have included the impact of GLP-1 on the reproductive system in the text and Table 2.
- The ileal brake activity of GLP-1 mentioned in line 135 might read better after line 123 when talking about its effects on gastric emptying as this mechanism is essential in reducing gastric emptying. It doesn’t fit well when discussing activity in the brainstem and hypothalamus unless an associated neural aspect is mentioned.
We have agreed with reviewer’s constructive suggestion. We have moved the sentence regarding the ileal brake activity of GLP-1 accordingly.
- Authors can also mention the primary source of endogenous GLP-1 in the brain that will help better understand effects of GLP-1R agonists in the brain.
We have added the sentence regarding the origin of endogenous GLP-1 in the brain as reviewer’s suggestion.
Text (2. GLP-1 analogs and physiological importance in humans):
“An endogenous GLP-1 is produced not only peripherally in the ileum, but also centrally in the brain, particularly in the area of nucleus of the solitary tract (NTS) of the hindbrain and the olfactory bulb, thereby, controls appetite and feeding behaviors [Daniels et al., 2019].” (Page 6, Line 149)
Reference:
- Daniels D, Mietlicki-Baase EG. Glucagon-Like Peptide 1 in the Brain: Where Is It Coming From, Where Is It Going? Diabetes. 2019;68(1):15-17.
- Paragraph starting with Line 164 can precede the paragraph starting with Line 148 to avoid discontinuity as immune cells incidentally come into the picture before the authors go back to the CNS.
We have agreed with reviewer’s suggestion. We have shifted the mentioned paragraph accordingly.
- in ‘GLP-1 analogs and cardiac glucose uptake’, the authors can add more information on myocardial glucose uptake, if it increases only in patients with decreased glucose uptake and touch upon the role of glucose transporters in mediating this effect in the heart.
We have discussed more about the role of GLP-1 analog on myocardial glucose uptake according to the reviewer’s suggestion as follows.
Text (3.2. GLP-1 analogs and myocardial glucose uptake):
“GLP-1-(7-36) amide was found to directly augment myocardial glucose uptake by increasing glucose transporters (GLUTs), GLUT-1 and GLUT-4, translocation in normal and I/R induced ischemic rat hearts in a similar pattern as insulin [Zhao et al., 2006]. Treatment with exendin-4 induced cardiac glucose uptake and ATP production via an increase of GLUT-1 translocation which was dependent to the expression of p38 mitogen-activated protein kinase (p38MAPK) in hypoxia/reoxygenation (H/R)‑induced injury in H9c2 cardiomyocyte cells [Lu et al., 2015]. However, clinical evidence reported that baseline functions of glucose transport and insulin resistance are the main determinant on the impact of GLP-1 analog on glucose transport and glucose uptake in cardiomyocytes during hyperglycemic states in patients with T2DM [Gejl et al., 2012] and during normo- or hypoglycemia in healthy subjects [Gejl et al., 2014].” (Page 9, Line 240)
Reference:
- Zhao T, Parikh P, Bhashyam S, Bolukoglu H, Poornima I, Shen YT, Shannon RP. Direct effects of glucagon-like peptide-1 on myocardial contractility and glucose uptake in normal and postischemic isolated rat hearts. J Pharmacol Exp Ther. 2006;317(3):1106-13.
- Lu K, Chang G, Ye L, Zhang P, Li Y, Zhang D. Protective effects of extendin‑4 on hypoxia/reoxygenation‑induced injury in H9c2 cells. Mol Med Rep. 2015;12(2):3007-16.
- Gejl M, Søndergaard HM, Stecher C, Bibby BM, Møller N, Bøtker HE, Hansen SB, Gjedde A, Rungby J, Brock B. Exenatide alters myocardial glucose transport and uptake depending on insulin resistance and increases myocardial blood flow in patients with type 2 diabetes. J Clin Endocrinol Metab. 2012;97(7):E1165-9.
- Gejl M, Lerche S, Mengel A, Møller N, Bibby BM, Smidt K, Brock B, Søndergaard H, Bøtker HE, Gjedde A, Holst JJ, Hansen SB, Rungby J. Influence of GLP-1 on myocardial glucose metabolism in healthy men during normo- or hypoglycemia. PLoS One. 2014;9(1):e83758.
- Line 219 seems quite repetitive as wide range of GLP-1 effects have already been mentioned in the article previously.
We have agreed with reviewer’s suggestion. We have revised by moving the text to Line 223 in order to make it more consistent.

Reviewer 3 Report
The review article by Pandey et al., that there are several lines of evidence which have revealed that GLP-1 mimetics possess cardiopro tective effects by various mechanisms related to cardiac contractility, cardiac glucose uptake, cardiac oxidative stress and ischemia/reperfusion injury, and mitochondrial homeostasis. In addition, upon binding to GLP-1R, GLP-1 and its analogs exert their exerts its effects via adenylyl cyclase mediated cAMP elevation, and subsequent protein kinase(s) triggering, which stimulate insulin release in conjunction with enhanced Ca2+ and ATP levels. They describe also that recent findings have suggested for additional downstream molecular pathways stirred by long-term GLP-1 analogs exposure which pave the way for the development of potential therapeutic molecules with longer lasting beneficial effects against diabetic cardiomyopathies. They provide a comprehensive overview of the recent advances in the understanding of GLP-1R-dependent and -independent actions of GLP-1 analogs in the protection of cardiomyopathies.
The review is well written and easy to follow. It is well structured and the figures are representative.
I have the following comments:
- The bibliography should be improved.
- There are some mistakes throughout the manuscript.
- The results found in humans versus animal models should be better discussed.
- There are some mistakes throughout the manuscript.
Author Response
Reviewer 3
The review article by Pandey et al., that there are several lines of evidence which have revealed that GLP-1 mimetics possess cardioprotective effects by various mechanisms related to cardiac contractility, cardiac glucose uptake, cardiac oxidative stress and ischemia/reperfusion injury, and mitochondrial homeostasis. In addition, upon binding to GLP-1R, GLP-1 and its analogs exert their effects via adenylyl cyclase mediated cAMP elevation, and subsequent protein kinase(s) triggering, which stimulate insulin release in conjunction with enhanced Ca2+ and ATP levels. They describe also that recent findings have suggested for additional downstream molecular pathways stirred by long-term GLP-1 analogs exposure which pave the way for the development of potential therapeutic molecules with longer lasting beneficial effects against diabetic cardiomyopathies. They provide a comprehensive overview of the recent advances in the understanding of GLP-1R-dependent and -independent actions of GLP-1 analogs in the protection of cardiomyopathies.
The review is well written and easy to follow. It is well structured, and the figures are representative. I have the following comments:
We would like to thank the reviewer for reviewing our manuscript and raising the insightful comments to our work. We have addressed the concerns raised by the reviewers by adding explanations to each query. Our point-by-point response is presented below, and we have highlighted (marked in yellow) where modifications to the manuscript text have been made.
- The bibliography should be improved.
Thanks for highlighting these issues. We have revised and updated some outdated references.
- There are some mistakes throughout the manuscript.
We thank the reviewer for raising these concerns. We have carefully revised the manuscript and also asked a native English speaker to check the language use and edit our manuscript in accordance with the reviewer’s suggestions. We hope the reviewer will acknowledge our effort to improve the usage of the English language in this manuscript.
- The results found in human versus animal models should be better discussed.
Although, current evidence has been shown that GLP-1R agonists express cardioprotective effects in both in vitro and in vivo studies, few clinical studies of these drugs have been investigated their efficacy in heart diseases, especially heart failure and cardiomyopathy. In addition, this drug class has not been approved by the US-FDA for use in patients with cardiomyopathy so far. For these reasons, we summarized the preclinical studies of GLP-1R agonists for cardioprotective effects both in vitro and in vivo in Table 3 and Table 4, respectively. Since, this review focuses on the molecular mechanisms and signaling pathways underlying cardioprotective effects of GLP-1 analogs, therefore, clinical outcomes from human trials have not been extensively discussed.
